# Preparation of Human Muscle Precursor Cells for the MyoGravity Project's Study of Cell Cultures in Experiment Units for Space Flight Purposes

Ester Sara Di Filippo [1,2] , Sara Chiappalupi [2,3], Michele Balsamo [4], Marco Vukich [4], Guglielmo Sorci [2,3,†] and Stefania Fulle [1,2,*,†]

1   Department of Neuroscience Imaging and Clinical Sciences, University "G. d'Annunzio" Chieti-Pescara, 66100 Chieti, Italy; es.difilippo@unich.it
2   Interuniversity Institute of Myology (IIM), 06132 Perugia, Italy; sara.chiappalupi@unipg.it (S.C.); guglielmo.sorci@unipg.it (G.S.)
3   Department Medicine and Surgery, University of Perugia, 06132 Perugia, Italy
4   Kayser Italia s.r.l., Via di Popogna, 501, 57128 Livorno, Italy; m.balsamo@kayser.it (M.B.); marco.vukich@ext.esa.int (M.V.)
*   Correspondence: stefania.fulle@unich.it
†   These authors contributed equally to this work.

**Featured Application: The results obtained by human muscle precursor cells cultured in a microgravity environment on board the ISS will be useful to design interventions to prevent or counteract muscle atrophy occurring during long-term space flights or on Earth in a plethora of conditions, including aging, disuse, denervation, malnutrition, pharmacological treatment with glucocorticoids, and cancer.**

**Abstract:** Long-time exposure to the microgravity conditions experienced during space flights induces alterations in the homeostasis of organs and tissues, including skeletal muscles, which undergo atrophy with the loss of mass and strength due to decreased size and altered composition of myofibers. Microgravity conditions can also affect the functionality of satellite cells, i.e., the adult stem cells providing the muscle precursors that are responsible for the growth and maintenance of muscle mass in adult life, as well as for muscle regeneration following a damage. The MyoGravity project, funded by Agenzia Spaziale Italiana (ASI), aimed to send human muscle precursor cells (huMPCs) on board the International Space Station (ISS) in order to study the effects of real microgravity on the differentiation capacity of this cell type. To this end, it was necessary to use a methodology to cultivate huMPCs inside dedicated space bioreactor devices (Experiment Units, EUs) specifically designed to cultivate cell cultures and perform scientific protocols in the space environment of the ISS. Here, we report the setting of several cell culture parameters to convert the EUs into suitable devices for biomedical experiments using huMPCs for space flight purposes.

**Keywords:** microgravity; human muscle precursor cells; satellite cells; muscle atrophy; space flights; international space station

## 1. Introduction

The microgravity experienced during space flights causes skeletal muscle atrophy, i.e., loss of muscle mass and strength, affecting extensor muscles more than flexor muscles, particularly during the first few weeks [1]. Muscle atrophy occurs when the protein degradation rate exceeds protein synthesis, and is induced in adult skeletal muscles by a variety of conditions, including malnutrition, denervation, cancer cachexia, heart failure and aging [2]. In microgravity conditions, the reduced protein synthesis is presumably triggered by the dramatically reduced load. Contractile proteins are lost out of proportion to other cellular proteins [1].

Besides affecting the protein turnover, microgravity may induce muscle atrophy by negatively affecting the biology and functionality of satellite cells (SCs) [3,4], a special cell type with a role in pre- and post-natal muscle growth. In adult skeletal muscles, SCs are quiescent and undifferentiated cells residing beneath the basal lamina and the sarcolemma [5], and they can undergo a rapid conversion from a quiescent to an activated state (myoblasts) in response to specific stimuli, including oxidative stress and muscle traumas. After activation, myoblasts proliferate extensively, and finally differentiate and fuse with each other to form myotubes, or with damaged myofibers and thus contribute to muscle regeneration following damage.

To further investigate the effects of microgravity on muscle stem cell biology, we planned to culture human muscle precursor cells (huMPCs) on board the International Space Station (ISS) with the use of specifically designed experiment units (EUs). EUs are electromechanical devices developed by Kayser Italia (Livorno, Italy; http://www.kayser.it/, accessed on 7 July 2022) allowing the autonomous execution of a scientific protocol in microgravity conditions. They are biocompatible devices providing the optimal conditions for the cells, including the possibility of supplying fresh mediums or specific reagents to culture, collect and preserve the samples at the end of the experiments. In particular, the experiment unit KEU-ST (also known as SPHINX or STROMA bioreactor) has been used over the past 10 years to cultivate a vast array of different cell cultures grown in a 2D environment on board the ISS [6–8]. However, very few studies exist regarding adherent human stem cells performed in space, and for the first time, thanks to the MyoGravity project funded by ASI, huMPCs have been sent on board the ISS.

The MyoGravity project was aimed at studying the effects of space flight on huMPC biology. Here we report on the preliminary tests performed to optimize several parameters to obtain the best culture conditions for huMPCs inside the EUs in order to grow, differentiate and store them on board the ISS.

## 2. Materials and Methods

### 2.1. Human Muscle Precursor Cells (huMPCs)

Biopsies from vastus lateralis (VL) muscles were obtained from healthy elderly people ($61 \pm 4$) by tiny percutaneous needle biopsy [9]. All subjects gave informed consent for inclusion before participating in the study. The study was conducted in accordance with the Declaration of Helsinki, and the protocol was approved by the Ethics Committee of Provinces of Chieti-Pescara (No. 04 of 25 February 2016 and No. 22 of 1 December 2016). To obtain huMPCs, the muscle biopsies were processed according to the procedure of Fulle [10]. Both the myogenic purity and the fusion index of the cultures were measured for each sample as described in Pietrangelo [11]. Briefly, myogenicity (%) was obtained by counting the huMPCs that were desmin-positive using a specific antibody (Dako REAL™ Detection System Peroxidase/DAB+, Rabbit/Mouse; DAKO, Glostrup, Denmark; #K5001), with respect to the total of cells present in the fields observed ($\geq$65% desmin-positive cells). The differentiation of huMPCs into myotubes was measured in terms of multinucleated cells ($\geq$3 nuclei per cell) positive for myosin heavy chain (MF20 antibody; Developmental Studies Hybridoma Bank, University of Iowa, Iowa City, IA, USA) after 7 days of differentiation. This is reported as the fusion index, i.e., the percentage of the number of nuclei into myotubes with respect to the number of total nuclei in the observed fields.

### 2.2. Cell Culture

Human muscle precursor cells (huMPCs) were cultured in a growth medium (GM) consisting of HAM's Nutrient Mixture F10 without L-Gluamine medium (Euroclone, Milan, Italy; #ECB7503L) supplemented with 20% Defined Fetal Bovine Serum, US Origin; (Euroclone, Milan, Italy; #CHA1111L), 1% Penicillin-Streptomycin Solution 100X (Euroclone, Milan, Italy; #ECB3001D) 0.1% Gentamicin 10 mg/ml sulphate (Euroclone, Milan, Italy; #ECM0011D) and 1% Stable Glutamine (200mM) (Euroclone, Milan, Italy; #ECB3004D) in an $H_2O$-saturated 5% $CO_2$ atmosphere at 37 °C. At a 60–70% confluence, the cells were

treated by adding trypsin-EDTA 1X in PBS w/o Calcium w/o Magnesium w/o Phenol Red (Euroclone, Milan, Italy; #ECB3052D) solution, harvested, centrifuged at 800 rpm for 5 min, suspended in fresh GM, counted in a Bürker chamber, and plated on plasticNunc™ Thermanox™ coverslips (10.5 × 22 mm, ThermoFisher Scientific, Waltham, MA, USA; #174934). Plastic coverslips were preferred to glass coverslips since these latter could break during launch and shipment to the ISS. We had already used plastic coverslips to cultivate muscle cells in both proliferation and differentiation conditions, obtaining good results [12]. The cell proliferation rate was evaluated as population doubling level (PDL) using the formula logN/ln2, where N represents the ratio between the number of cells that were detached after 10 days of culture in GM and the number of cells that were initially seeded. To evaluate cell survival, huMPCs were trypsinized, suspended in PBS, and added with trypan blue 0.4% solution (Sigma-Aldrich, Milan, Italy; #T8154) (1:1 *v/v*). Cell viability was determined as the percentage of trypan blue-negative cells with respect to the total amount cells.

To induce differentiation into myotubes, huMPCs were added to a differentiation medium (DM), composed of Dulbecco's Modified Eagle's Medium high-glucose (DMEM) Euroclone, Milan, Italy; #ECB7501L) supplemented with 5% heat-inactivated (56 °C, 30 min) horse serum (Euroclone, Milan, Italy; #ECS0091L), 50 µg/mL Gentamicin 10 mg/ml sulphate, 1% Stable Glutamine (200 mM), 1% Penicillin-Streptomycin Solution 100X, 10 µg/mL insulin solution from bovine pancreas (Sigma-Aldrich, Milan, Italy; #I0516-5ml) and 100 µg/mL of apo-transferrin human (Sigma-Aldrich, Milan, Italy; #T2036). We followed the differentiation process until 7 days.

### 2.3. Experimental Hardware (EHs)

Each KEU-ST EU (also called STROMA or SPHINX bioreactors) developed by Kayser Italia Srl (Livorno, Italy) is composed of a brick of biological compatible plastic (PEEK) with a cell culture chamber, five cylindrical reservoirs to store media and chemicals, and a fluidic path that allows for fluid displacement from the reservoirs towards the culture chamber and for the recovering of the exhausted media. The culture chamber is designed to accommodate cells cultured in a monolayer on a 2.3 cm$^2$ Thermanox™ coverslip with around 1.1 mL of culture medium. Fluid displacement is actuated by a piston that injects fresh fluids into the culture chamber, moving the wasted fluids on the rear side of the piston [13]. The solutions contained inside the EUs reservoirs were different depending on whether the sample has to be processed or not for RNA extraction.

### 2.4. Cell Growth Inside the Experiment Units

The culture chamber in the EU is designed to accommodate cells cultured in a monolayer on 2.3 cm$^2$ in around 1.1 mL of culture medium. Before the integration into the EUs, we seeded huMPCs on the coverslips in GM and cultured them for 3 days at 37 °C and 5% $CO_2$. To avoid the cells migrating out the coverslip by sticking to the plate, they were drop seeded on the slide. After adhesion to the slide (approx. 12 h later) GM was added and the cells were left for another 48 h in GM in a 37 °C and 5% $CO_2$ incubator before being integrated into the EUs. To identify the best coating agent the coverslips were pre-coated with different matrices: ECL cell attachment matrix (entactin-collagen IV-laminin) (Sigma-Aldrich, Milan, Italy; #08-110), Gelatin solution (Sigma-Aldrich, Milan, Italy; #G1393-20ML), Poly-D-lysine hydrobromide (Sigma-Aldrich, Milan, Italy; #P6407-5MG), or Human recombinant Laminin-521 (BioLamina, Sundbyberg, Sweden; #LN521). Coverslips with ECL or Laminin-521 were pre-coated overnight at 4 °C. Poly-D-lysine hydrobromide solution was applied to the coverslip for 5 min, removed through aspiration, and the coverslip surface thoroughly rinsed. Before use, the coverslip was allowed to dry for 2 h. Gelatin solution was placed to cover the coverslip, aspirated, and then left for 10 min at 37 °C. We seeded $5.0 \times 10^5$ cells on each coverslip in GM for 24 h and observed the coverslips by an inverted microscope.

To evaluate cell survival in the EUs, where no gas exchange is permitted, we seeded huMPCs at $2.0 \times 10^4$ or $4.0 \times 10^4$ cells per coverslip in GM or DM supplemented with 20, 25 or 40 mM HEPES (Sigma-Aldrich, Milan, Italy; #H4034). To determine the best cell density, huMPCs were seeded at $1.0 \times 10^4$, $2.0 \times 10^4$ or $5.0 \times 10^4$ cells per coverslip for the proliferation condition, and $2.5 \times 10^4$, $4.0 \times 10^4$, $5.0 \times 10^4$ or $7.5 \times 10^4$ cells per coverslip for the differentiation condition.

### 2.5. RNA Extraction and Quantification

RNA was extracted from the huMPCs cultured on plates or in EUs, in growth and differentiation conditions. For RNA extraction from huMPCs cultured in the EUs in proliferation conditions, the reservoirs were filled following the order: Reservoir 1, GM (20 mM HEPES); Reservoir 2, GM (20 mM HEPES); Reservoir 3, HAM's F10 (20 mM HEPES); Reservoir 4, RNAlater Solution (Invitrogen, ThermoFisher Scientific, Waltham, MA, USA; #AM7020); Reservoir 5, empty; Culture chamber, GM (20 mM HEPES). For RNA extraction from huMPCs cultured in the EUs in differentiation conditions the reservoirs were filled following the order: Reservoir 1, DM (20 mM HEPES); Reservoir 2, DMEM (20 mM HEPES); Reservoir 3, RNAlater Solution; Reservoir 4, empty; Reservoir 5, empty; Culture chamber, GM (20 mM HEPES). To reduce the detachment of cells from coverslip, HAM's F10 or DMEM were used instead of PBS as the washing media before the addition of RNAlater Solution. The medium of the cultures in the proliferation conditions was changed with the following timing: 1st change (GM) after 5 days from the start of the experiment, 2nd change (GM) after other 3 days, 3rd change (HAM's F-10) after a further 3 days, 4th change (RNAlater) a5 min after the 3rd change. For cultures in differentiation conditions, the medium changes occurred as follows: 1st change (DM) 5 days after the start of the experiment, 2nd change (DMEM) after an additional 3 days, 3rd change (RNAlater) 5 min after the 2nd change. For RNA extraction from huMPCs cultured on plates, the same changes of medium were performed as those used with the EUs (with the addition of 20 mM HEPES).

To identify the proper RNA isolation kit, cells were seeded on Thermanox™ coverslips ($4.0 \times 10^4$ cells per coverslip) and grown in DM for 7 days on plates or inside the EUs, and then pre-treated with RNAlater, a nontoxic reagent able to permeate cells and stabilize cellular RNAs. The RNA was extracted with the following kits according to the manufacturer's instructions: (i) mirVana miRNA Isolation kit (Invitrogen by ThermoFisher Scientific, Waltham, MA, USA; #AM1560); (ii) Purelink RNA Mini Kits (Ambion by Life Technologies, Monza, Italy; #12183018A) followed by PureLink miRNA Isolation Kit (Invitrogen by Life Technology, Monza, Italy; #K1570-01), which gave the option of isolating RNA and microRNA separately; (iii) MiRNeasy Micro Kit (Qiagen, Milan, Italy; #217084). The RNA obtained was quantified using a NanoDrop™ spectrophotometer and the 260/280 ratios were evaluated.

### 2.6. Quantitative Real-Time PCR for Myogenic Regulatory Factor

Retro-transcription and real-time PCR were performed according to the Applied Biosystem's High Capacity cDNA Reverse Transcription kit (Applied Biosystem, Life Technologies, Monza, Italy; #4368814). Quantitative real-time PCR was performed using the TaqMan probes and the specific TaqMan Universal Master Mix II, no UNG (Applied Biosystem, Life Technologies, Monza, Italy; #4440040), in 96-well plates. We evaluated the expression of the following human genes: paired box 7 (*PAX7*) (Hs00242962_m1; #4331182); Myogenic Differentiation 1 (*MYOD1*) (Hs00159528_m1; #4331182), myogenin (*MYOG*) (#4331182, Hs 01072232_m1). Glyceraldehyde-3-phosphate dehydrogenase (*GAPDH*) (Hs99999905_m1; #4331182) was used as the internal control, and the data are shown as difference in cycle threshold ($\Delta$Ct). An Applied Biosystems Prism 7900HT Sequence Detection System was used, with the Sequence Detector Software (SDS version 2.0; Applied Biosystems) [14]. The more that the values are positive, the more the genes are downregulated, and vice versa [15].

### 2.7. MicroRNA Expression Profile

Retro-transcription and quantitative real-time PCR (qRT-PCR) were carried out according to the Applied Biosystems TaqMan miRNA assay kit protocols. Briefly, the retro-transcription involved 20 ng of a small RNA, as the "stem loop" primer specific for each miRNA, dNTPs and inverse transcriptase RNAse inhibitors, according to the Applied Biosystems™ High-Capacity cDNA Reverse Transcription kit, using a Thermocycler (Applied Biosystem, Life Technologies, Monza, Italy) (30 min at 16 °C, 30 min at 42 °C, 5 min at 85 °C, then at 4 °C). Then, qRT-PCR analysis for the miRNA expression levels was performed using the TaqMan probes and the specific TaqMan® Universal Master Mix II, no UNG in 96-well plates with a PRISM 7900 HT Sequence Detection System (Applied Biosystems) in triplicate. The specific miRNA probes used were: has-miR-1 (#002222); has-miR-133a (#002246); has-miR-133b (#002247); has-miR-206 (#000510); has-miR-16-5p (#000391), all from Applied Biosystems. MiR-16 was used as the endogenous control. The relative quantification of the miRNA targets was carried out using the $\Delta$Ct formula (Ct $_{\text{miRNA of interest}}$ − Ct $_{\text{miR-16}}$), according to the Ct method. The more values that were positive, the more the miRNA were downregulated, and vice versa [16].

### 2.8. Statistical Analysis

The statistical analysis was carried out using GraphPad Prism Software, version 9.3.1 (GraphPad Software, La Jolla, CA, USA). The data are reported as the mean $\pm$ SEM, performing unpaired *t*-tests to reveal the statistical differences.

## 3. Results

### 3.1. Cell Culture Optimization into the EUs

In view of carrying out experiments with huMPCs on board the ISS with the use of the EUs, we performed preliminary tests to set up the following parameters: (i) the type of coating to apply of Thermanox™ plastic coverslips to obtain the best cell adhesion; (ii) the best HEPES concentration to buffer the medium, since the EUs are closed systems that prevent any gas exchange; (iii) the resilience of cells to the temperature profile foreseen during the upload on the ISS; (iv) the cell density to place on the coverslips to obtain an efficacious myogenic differentiation; and (v) the RNA extraction kit to be used to obtain the best yield in terms of quantity and purity.

#### 3.1.1. Type of Coating

We tested several kinds of cellular coating (Figure 1A,B) to find the most efficient one in improving the adhesion of huMPCs on Thermanox™ coverslips: (i) ECL cell attachment matrix (entactin-collagen IV-laminin); (ii) gelatin solution; (iii) Poly-D-Lysine hydrobromide; and (iv) human recombinant Laminin-521.

We found that Laminin-521 gave the best results in terms of cell adhesion. Indeed, in the presence of Laminin-521, huMPCs appeared adherent and uniformly distributed on the coverslip with almost no cells found suspended in the culture medium (Figure 1A). This result was confirmed by counting the cell numbers per field, which showed a significantly higher number of adherent cells in the presence of Laminin-521 compared to the other coating media tested (Figure 1B).

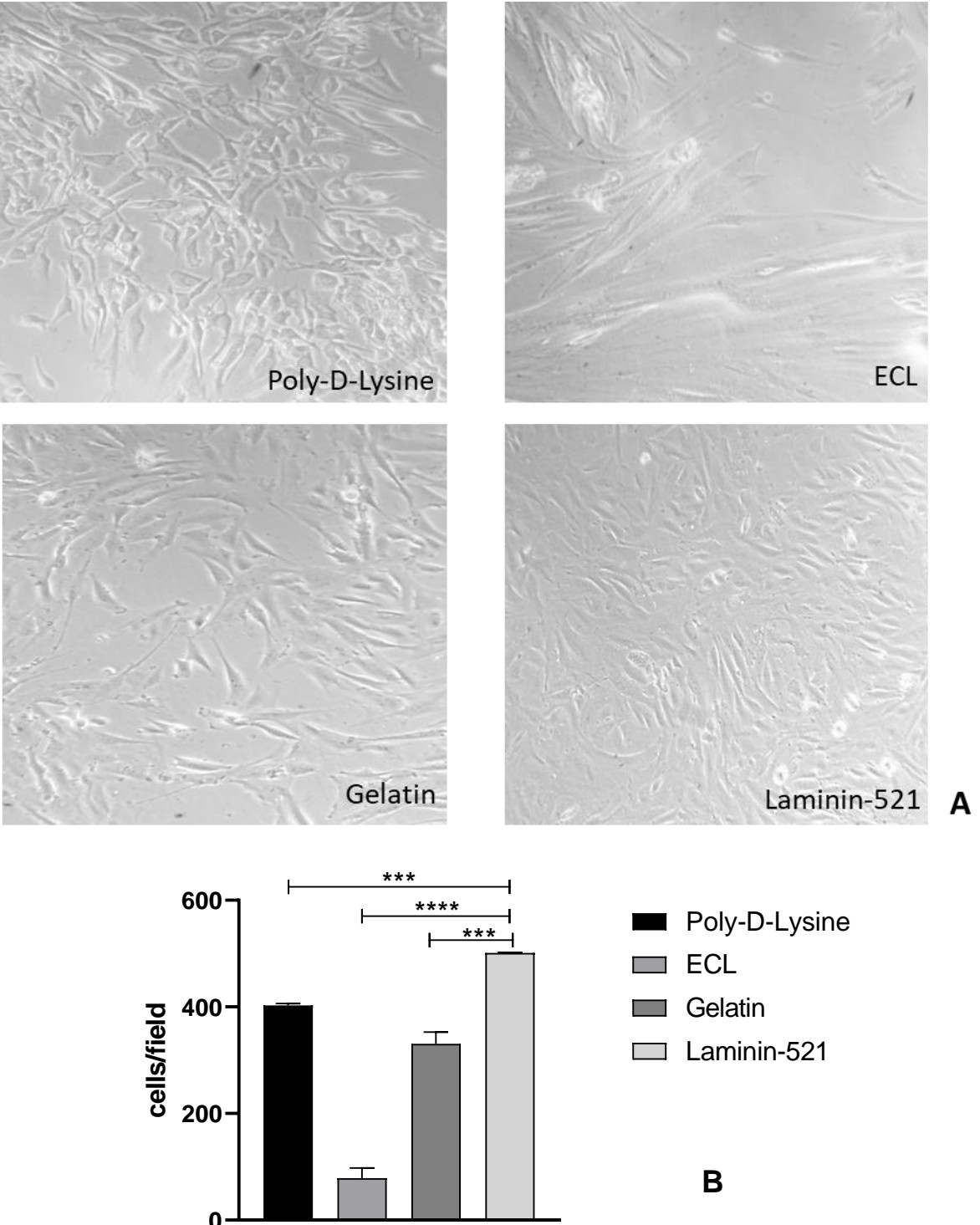

**Figure 1.** (**A**) Phase contrast representative images of huMPCs seeded in GM ($5.0 \times 10^5$ cells per coverslip) on Thermanox™ coverslips pre-coated with poly-D-lysine hydrobromide, ECL, gelatin or Laminin-521. Cells were cultured for 24 h in standard conditions and viewed by an inverted microscope. Magnification, 20×. (**B**) Graph with the results of cell counts in the presence of the diverse coating media used. At least 500 cells were counted in 10 different randomly selected fields (*** $p < 0.001$, **** $p < 0.0001$).

### 3.1.2. HEPES Concentration

Since the EUs do not allow for gas exchange, the cell culture medium needed to be buffered to maintain the required physiological pH. Thus, we seeded huMPCs in GM or DM supplemented with different HEPES concentrations, according to other studies [17–19]. Based on the proliferative and survival rates, the best HEPES concentration for huMPC culture in the EUs resulted in 20 mM, since it guaranteed a cell viability comparable to that obtained in standard culture conditions in plates (Table 1).

**Table 1.** Cell survival and proliferation rates in dependence on HEPES concentration. Percentage cell survival and proliferation of huMPCs cultured in EUs in the presence of the indicated HEPES concentrations with respect to huMPCs cultured in parallel in plates in standard medium conditions.

| HEPES (mM) | 20 | 25 | 40 |
|---|---|---|---|
| Cell survival (%) | 98.3 ± 3.2 | 95.1 ± 3.5 | 70.7 ± 2.6 |
| Cell proliferation (%) | 97.5 ± 3.9 | 83.4 ± 4.2 | 25.2 ± 3.7 |

### 3.1.3. Cell Resilience to Temperature

For the execution of the MyoGravity project, from handover to NASA integrators to the installation into KUBIK, the EUs would be assembled with control electronics integrated inside the KIC-SL containers (Kayser Italia Containers-Single Level) and then, for the upload on board the launcher, placed inside the BIOKIT transportation container, which is a soft pouch containing phase change materials (PCM bricks) preheated at 27 °C, and is used to control temperature. Once aboard the ISS, the KIC-SL containers would be inserted inside the KUBIK incubator developed by European Space Agency (ESA) and operative in the ISS Columbus module, set at 37 °C. Five days was the maximum time expected to reach the ISS, during which significant variations of temperature were possible. In fact, during the flight phases the temperature control was based on a passive system rather than an active incubator. Using data from previous expeditions, an expected temperature (T) profile in the worst-case scenario was defined in which 27.6 °C was the lowest T value (Figure 2).

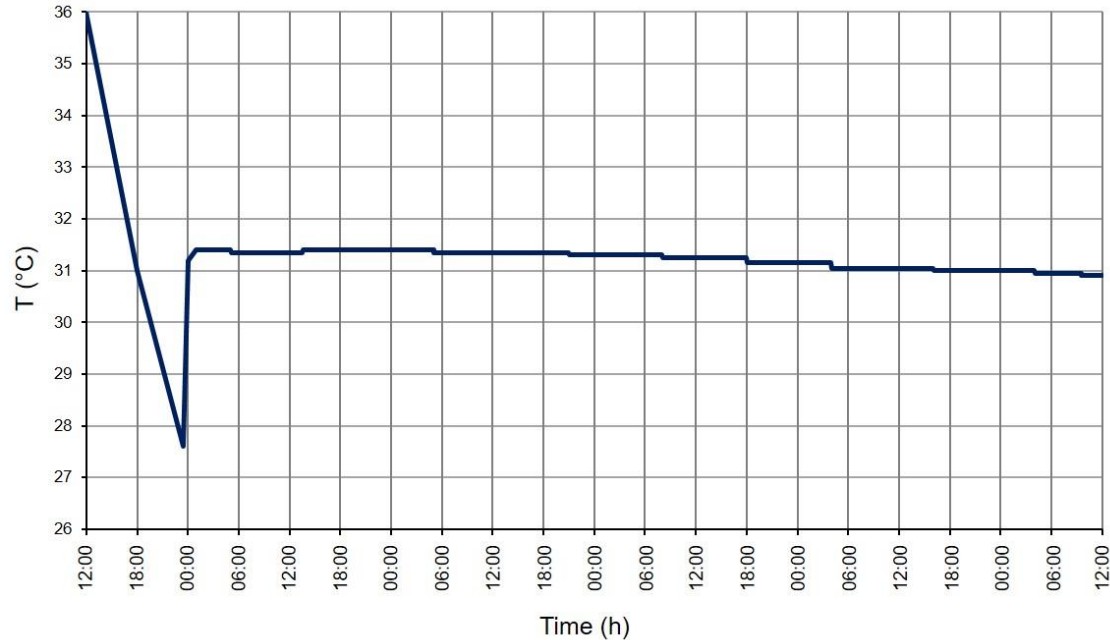

**Figure 2.** The T (°C) profile over time (h) in the worst-case scenario during the transport of EUs inside the KIC-SL containers and inside the TBC transportation container until the integration into the KUBIK on board the ISS.

Thus, we performed tests to evaluate the survival and proliferation rates of huMPCs at the T characterizing the shipment from handover to the integration into the KUBIK on board the ISS. Based on the worst-case scenario T profile (Figure 2), we tested the cell resilience at 25, 27 and 30 °C. The results obtained showed that the lowest temperature required to ensure the normal survival of huMPCs for 5 days was 27 °C, as evaluated by vital cell count (Table 2).

**Table 2.** Cell resilience to temperature. Cell survival percentage of huMPCs cultured in the EUs at the indicated T for 5 days with respect to huMPCs cultured in parallel at 37 °C in plates (control).

| Temperature (°C) | 25 | 27 | 30 |
|---|---|---|---|
| Cell survival (% of control) | $30.5 \pm 4.2$ | $98.2 \pm 6.2$ | $98.6 \pm 6.5$ |

### 3.1.4. Cell Density to Plate

Considering that the period from handover to the installation into the KUBIK could last 5 days (worst-case scenario) without any possibility to change the culture medium, we seeded huMPCs on Thermanox$^{TM}$ coverslips at different densities to establish the appropriate one to be used to study cell proliferation or cell differentiation avoiding scarce or excessive cell confluence. We seeded huMPCs at $1.0–5.0 \times 10^4$ or $2.5–7.5 \times 10^4$ cells per coverslip for the proliferation and differentiation condition, respectively After 3 days in GM, cells were cultured for a further 5 days at T mimicking the T profile reported in Figure 2, and then cultivated for additional 3 days in GM or in DM to induce differentiation. We found that $2.0 \times 10^4$ and $4.0 \times 10^4$ cells per coverslip were the best plating densities for cell proliferation and cell differentiation purposes on board the ISS, respectively (data not shown).

Therefore, we evaluated huMPCs seeded at the identified densities and cultured in the above reported conditions on plates or in EUs and found that they displayed similar proliferation and differentiation capabilities (Figure 3).

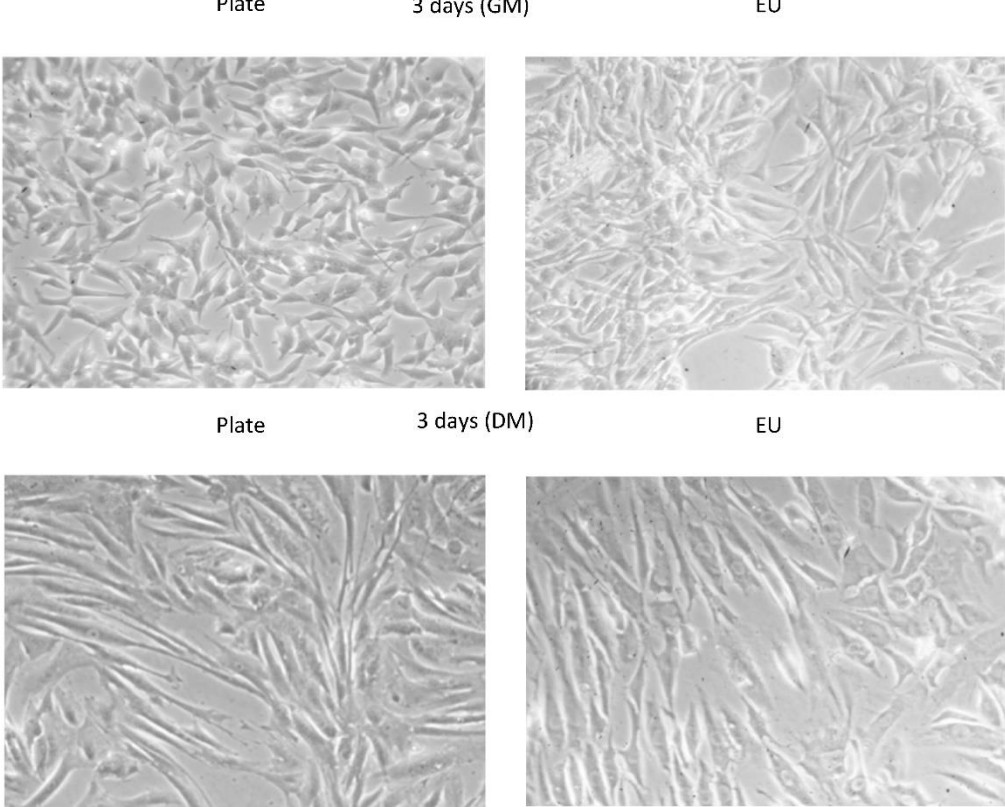

**Figure 3.** Myoblasts and myotubes after 3 days of culture in proliferation (GM) or differentiation (DM) conditions as viewed by phase-contrast microscopy. Magnification, 20×.

We calculated the proliferation rate expressed as population doubling level (PDL) after 10 days of culture in GM, and the capability to differentiate expressed as ability to fuse (Fusion Index, FI%) after 7 days in DM (Table 3). We found that huMPCs showed similar PDL (4.0 vs. 3.8) and FI (55.2% vs. 57.0%) when cultured on plates or in EUs.

**Table 3.** Population doubling level (PDL) and fusion index (FI). The proliferation rate was calculated as the PDL after 10 days of culture. The ability to fuse was calculated as FI (%) after 7 days of differentiation.

| Conditions | Plate | EU |
|---|---|---|
| PDL | $4.0 \pm 0.2$ | $3.8 \pm 0.1$ |
| FI (%) | $55.2 \pm 5.1$ | $57.0 \pm 2.3$ |

3.1.5. RNA Extraction

Finally, we moved to the identification of the most appropriate RNA extraction kit for huMPCs, in order to get the best RNA yield in terms of quantity and purity. The following commercial kits, which also allow for miRNA extraction, were evaluated: (i) mirVana miRNA Isolation kit; (ii) PureLink RNA Mini Kit followed by PureLink miRNA Isolation Kit; and (iii) miRNeasy Micro Kit. The results showed that the optimal RNA extraction, in terms of yield and 260/280 ratio, was obtained with miRNeasy Micro Kit, both for on plates and in EUs. Indeed, we obtained an average of 72.5 and 78.1 ng/µL of total RNA per coverslip, and 1.98 and 1.90 as 260/280 ratios for huMPCs cultured on plates or in EUs, respectively (Table 4).

**Table 4.** RNA quantification from single Thermanox^TM coverslip cultured in plates or EUs. Total RNA or microRNAs isolated using the indicated kits from huMPCs seeded on Thermanox™ coverslips and cultured either on plates or inside the EUs. Quantification was performed by a NanoDrop™ spectrophotometer. The average 260/280 ratios are reported.

| RNA Isolation Kit | Yield on Plates ng/µL (260/280 Ratio) | Yield inside the EUs ng/µL (260/280 Ratio) |
|---|---|---|
| mirVana miRNA | total RNA 18.6 (2.05) | total RNA 29.5 (1.54) |
| PureLink miRNA PureLink RNA Mini Kit | microRNAs 3.0 (1.96) total RNA 16.8 (2.06) | microRNAs 3.2 (1.90) total RNA 18.9 (2.13) |
| miRNeasy Micro Kit | total RNA 72.5 (1.98) | total RNA 78.1 (1.90) |

*3.2. Expression Profile of Muscle-Specific Factors and MicroRNAs in the EUs*

We selected the muscle-specific factors and microRNAs with known expressions in huMPCs under both proliferation and differentiation conditions in order to evaluate if culturing in the EUs affects huMPC physiological behavior.

3.2.1. Expression Profile of Muscle-Specific Factors

We analyzed the expression of the muscle-specific factors, *PAX7*, *MYOD1* and *MYOG* in huMPCs cultured on plates or in EUs under standard gravity and in proliferation or differentiation conditions by qRT-PCR. We found that the selected genes showed a similar expression trend on plates and in EUs (Figure 4). In particular *PAX7*, a gene characterizing quiescent satellite cells and proliferating myoblasts, and *MYOD1*, a gene implicated in the proliferative boost required to commit and sustain differentiation, appeared similarly expressed in huMPCs cultured on plates and in EUs. *MYOD1* also appeared upregulated and similarly expressed on plates and in EUs in differentiation conditions. *MYOG*, a marker of myogenic terminal differentiation, was found upregulated in huMPCs cultured in DM, as expected, both on plates and in EUs (Figure 4). These data suggested that the use of the EU system does not affect the physiological expression of muscle-specific factors in these cells.

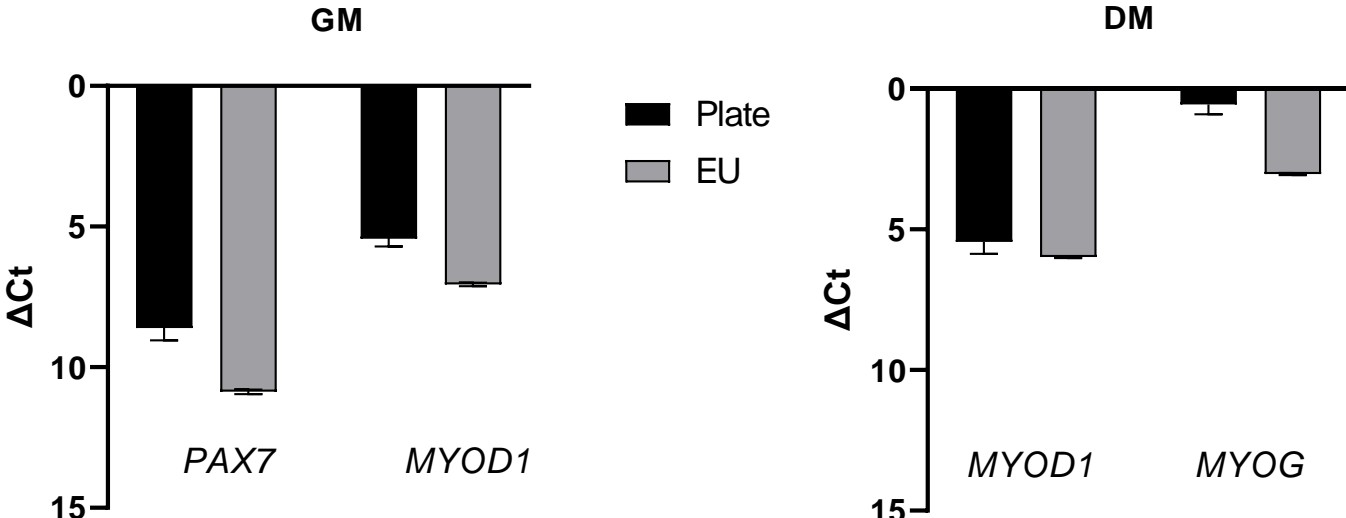

**Figure 4.** Expression of muscle-specific factors in huMPCs. The graphs show the relative expression of *PAX7*, *MYOD1*, and *MYOG* genes with respect to *GAPDH* in huMPCs cultured on plates or in EUs in both proliferation (GM) or differentiation (DM) conditions, as evaluated by qRT-PCR. The data are the mean $\pm$ SD of three independent experiments, each performed in triplicate.

### 3.2.2. Expression of myomiRNAs

The expression levels of the muscle-specific miRNAs (myomiRNAs), miR-1, miR-133a, miR-133b and miR-206 were analyzed in huMPCs during proliferation and after 7 days of differentiation on plates or in EUs under standard gravity conditions. The investigated myomiRNAs showed the same expression trend in cells cultured on plates and in EUs. Indeed, miR-1, miR-133a, miR-133b and miR-206, whose expression has been reported to increase during myogenic differentiation [15], were all up-regulated in huMPCs in differentiation compared to proliferation conditions independently from the culture device used (Figure 5), suggesting that culturing huMPCs in the EUs does not affect the expression of muscle-related microRNAs.

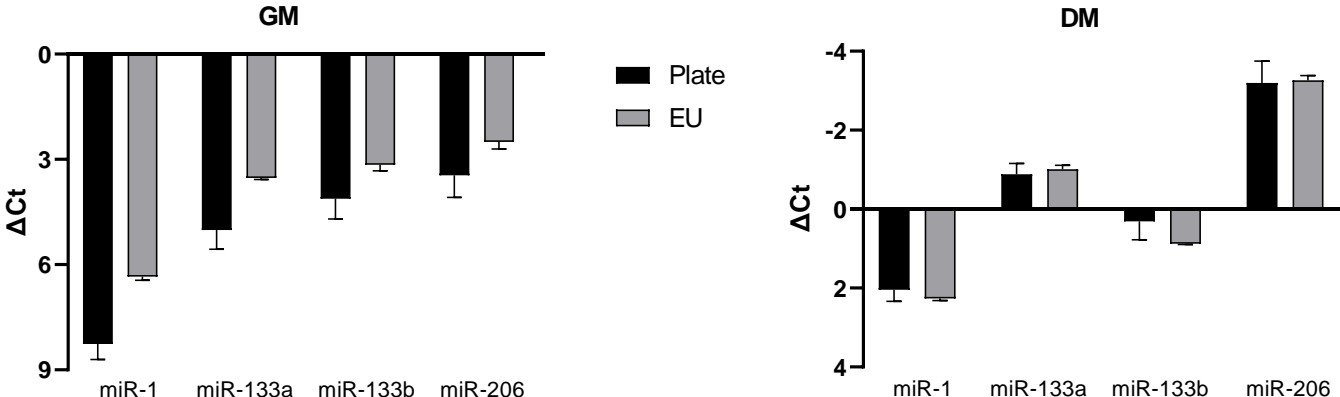

**Figure 5.** MicroRNA expression profile in huMPCs. The graphs show the relative expression of miR-1, miR-133a, miR-133b and miR-206 in both proliferation (GM) and differentiation (DM) conditions, as evaluated by qRT-PCR. The cells were cultivated on plates and in EUs. The data are the mean $\pm$ SD of three independent experiments, each performed in triplicate.

Table 5 resumes the identified optimal parameters in order to send and cultivate huMPCs on board the ISS.

**Table 5.** Selected parameters to be used with huMPCs to be sent on board the ISS.

| | |
|---|---|
| **Coating of Thermanox™** | Human Recombinant Laminin-521 (BioLamina) |
| **HEPES concentration** | 20 mM |
| **Lowest temperature tolerated** | 27 °C |
| **Plating densities for proliferation purposes** | $2.0 \times 10^4$ cells per coverslip |
| **Plating densities for differentiation purposes** | $4.0 \times 10^4$ cells per coverslip |
| **RNA extraction** | miRNeasy Micro Kit (Qiagen) |

## 4. Discussion

The weightlessness experienced as a result of the lower levels of gravity on board the ISS has negative impacts on astronauts' organs and tissues. Skeletal muscle is one of the most affected tissues during long stays in space, undergoing atrophy as a result of loss of mass due to a decrease of muscle fiber size [20] and alterations in the myofiber composition [21]. The adaptive pathophysiological changes that astronauts experience during space flights are similar to an accelerated aging process and to certain diseases [22]. While the average loss of muscle tissue with age is about 1% per year starting from peak development at about 24–30 years of age, with significant cumulative loss with advancing years, astronauts in space lose about 1% of muscle mass a month.

The muscle atrophy, which occurs over a period of weeks to months of exposure to weightlessness with an overall 7–20% size decrease [23], can be only partially contained with adequate in-flight physical exercise and, in due course, is mostly reversible after landing. Indeed, despite exercise countermeasures [24], the crewmembers experienced muscle atrophy, particularly in leg muscles, in long-duration (6 months) missions [25,26].

Investigating the molecular mechanisms of muscle atrophy induced by microgravity is essential for establishing clinical approaches and interventions to counteract this process and promote the growth and maintenance of muscle mass, not only during long-term space flights, but also in the atrophying processes linked to aging and several diseases. An advantage of the studies performed in microgravity conditions is that microgravity induces muscle atrophy much faster than aging. SCs, and the MPCs they provide, play a fundamental role in the maintenance of muscle mass in adulthood and in the muscle regeneration process following damage [15]. Muscle atrophy occurring during space flights is thought to be related, at least in part, to a deficit of these cells.

Various studies have shown that cells grown in simulated microgravity (with the use of a random positioning machine) undergo morphological and molecular alterations [4,27,28]. However, very few studies were carried out on cells under real microgravity on board the ISS. These include the MyoGravity project whose aim was to study the functional alterations induced by real microgravity in huMPCs. This kind of study requires the use of dedicated devices in which several culture parameters have to be adapted, especially in the case of adherent cell types. We used specifically designed EUs (Kayser Italia, Livorno, Italy) and defined the optimal culture conditions in terms of coverslip coating, HEPES concentration in the culture medium, and cell density at plating. Moreover, we evaluated the resilience of huMPCs to T variations occurring during the rocket launch and flight to the ISS, and identified the best kit to isolate the RNAs from huMPCs inside the EUs.

Importantly, we performed several analyses showing that the use of the set parameters allows culturing huMPCs in the EUs without significantly affecting their proliferation rate and differentiation extent in comparison with standard cultivation on plates. Indeed, the developed protocol reduces at the minimum the differences between huMPC cultures on plates and in EUs, in terms of population doubling level, fusion index, expression of muscle specific factors (*PAX7*, *MYOD1* and *MYOG*), and myomiRNAs (miR-1, miR-133a, miR-133b and miR-206), thus suggesting that the EUs are useful devices to study the biological responses of huMPCs to real microgravity on board the ISS.

The knowledge of the microgravity-induced changes in the properties of huMPCs, combined with the changes in their ability to respond to certain extracellular factors, will help to design future interventions to promote the maintenance of muscle mass during long-term space missions. Moreover, the knowledge of the microgravity-induced effects on huMPCs might be useful to better understand the molecular mechanisms underlying diseases characterized by muscle atrophy.

**Author Contributions:** Conceptualization, G.S. and S.F.; methodology, E.S.D.F. and S.C.; validation, G.S., M.B., M.V. and S.F.; formal analysis, E.S.D.F. and S.F.; investigation, E.S.D.F., S.C., G.S. and S.F.; resources, E.S.D.F. and M.V.; writing—original draft preparation, E.S.D.F. and S.F.; writing—review and editing, E.S.D.F., S.C., M.B., M.V., G.S. and S.F.; visualization, E.S.D.F., S.C., G.S. and S.F.; supervision, S.F.; project administration, S.F.; funding acquisition, S.F. All authors have read and agreed to the published version of the manuscript.

**Funding:** This research was funded by Agenzia Spaziale Italiana (ASI), MyoGravity project to S.F. (prot ASI DC-MIC-2012-24—contract 2016-4-U.0).

**Institutional Review Board Statement:** The study was conducted in accordance with the Declaration of Helsinki and approved by the Ethics Committee of the Provinces of Chieti and Pescara (protocol codes n. 4 25/02/2016 and n. 22 01/12/2016).

**Informed Consent Statement:** Informed consent was obtained from all subjects involved in the study.

**Data Availability Statement:** Not applicable.

**Conflicts of Interest:** The authors declare no conflict of interest.

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
