# Peer review of "Preparation of Human Muscle Precursor Cells for the MyoGravity Project’s Study of Cell Cultures in Experiment Units for Space Flight Purposes"

_applsci, doi:10.3390/app12147013_

Round 1

Reviewer 1 Report

The goal of this paper to is establish parameters for the use of Experimental Units to grow huMPCs on the ISS. This is a description of the optimization process performed on Earth. Because this is essentially a description of an earth-bound optimization process it is not extremely high impact; however, this information would be critical for other researchers to want to use the EU modules at a later date. I feel that the science is robust but have a few concerns: 

1. Why choose to do a media change at 5/3/3 days? You have an extra chamber in all of your experiments. Would it not be beneficial to do a media change every 3 days? Are you assuming that those initial 5 days are the 5 days of transport? 

2. I was initially confused as to why you selected the specific genes and and microRNAs that you did. You do eventually explain why some of the genes are of interest at the end of the results but I recommend that you move that information to the methods. A justification of the microRNAs selected would be useful. 

3. You mention that you perform measures to identify the percentage of cells that remain adhered versus those that become suspended.  Yet in Figure 1 you only show qualitative images of the disc surface. Could you provide the quantitative data? Is it showing numerical significance?

4. In Figure 2 we could not see the bottom of the plot. 

5. Your results section contains so much of the methods. You describe the entire bulk of the experiment in the Results section. This is not a major issue, but did make things difficult to understand for a bit. Maybe you need to move the bulk of the results section to the methods. 

6. In figures 4 and 5 you are having us compare the effects on the plate and the EY; however, you organize your plots such that each plot compares the RNA signals instead of the Plate/EU. I think it would be easier for the reader if you had plots of each gene showing Plate vs EU instead of the way it currently stands. 

Author Response

We wish to thank the Reviewer for her/his overall positive evaluation of the manuscript, and for her/his suggestions that enabled us to improve the manuscript. Here is our point-to-point reply to the Reviewer comments.

  1. Why choose to do a media change at 5/3/3 days? You have an extra chamber in all of your experiments. Would it not be beneficial to do a media change every 3 days? Are you assuming that those initial 5 days are the 5 days of transport?

Actually, we had to consider the worst-case scenario. As reported in session 3.1.4. (Cell density to plate), the period from handover to the installation into the KUBIK could last up to 5 days, with no possibility to change the culture medium by the automation system. This forced us to perform experiments using a 5-day incubation of the cells. We have stressed this concept in the revised version of the manuscript (Session 3.1.4.” Considering that the period from handover to the installation into the KUBIK could last 5 days (worst-case scenario) without any possibility to change the culture medium,...”).

  1. I was initially confused as to why you selected the specific genes and microRNAs that you did. You do eventually explain why some of the genes are of interest at the end of the results but I recommend that you move that information to the methods. A justification of the microRNAs selected would be useful.

We have added in the text the sentence, “We selected factors and microRNAs with known expression in huMPCs in proliferation and differentiation conditions in order to evaluate if culturing in the EUs affects huMPC physiological behavior” (Section 3.2. Expression profile of muscle-specific factors and microRNAs in the EUs).

  1. You mention that you perform measures to identify the percentage of cells that remain adhered versus those that become suspended. Yet in Figure 1 you only show qualitative images of the disc surface. Could you provide the quantitative data? Is it showing numerical significance?

We have improved Figure 1 by adding a graph with the results of cell counts in the presence of the diverse coating media used. These cell counts confirm a significantly higher number of adherent cells in the presence of Laminin-521 compared to the other coating media tested.

  1. In Figure 2 we could not see the bottom of the plot.

We apologize for this. We have replaced the figure.

  1. Your results section contains so much of the methods. You describe the entire bulk of the experiment in the Results section. This is not a major issue, but did make things difficult to understand for a bit. Maybe you need to move the bulk of the results section to the methods.

We thank the Reviewer for this suggestion. We have moved part of the results in the Methods section. The modifications are evidenced in red and barred.

  1. In figures 4 and 5 you are having us compare the effects on the plate and the EY; however, you organize your plots such that each plot compares the RNA signals instead of the Plate/EU. I think it would be easier for the reader if you had plots of each gene showing Plate vs EU instead of the way it currently stands.

Based on the Reviewer’s suggestion, we have reorganized the graphs in Figures 4 and 5 to allow easy comparison between huMPCs cultured in plates and EUs in GM or DM conditions. 

Reviewer 2 Report

The authors describe suitable methods to culture huMPCs that enables the optimum survival and proliferation cells in the challenging environments of spaceflight including ground transport / variable temperature incubations / and ISS spaceflight, in order to better ascertain the changes that microgravity induces on these critical cells.

The well-described results and discussion provide suitable evidence that we may use these methods into the future for studying spaceflight induced pathogenesis. The language is acceptable, and I detect no glaring issues

My only minor comment is the combination of graphs (figure 4, figure 5) to reduce whitespace and improve comparison between GM/DM and Plate/EU.

Increase in contrast of images for figure 2, and in greyscale, would improve visibility of cells (like figure 3)

Finally, it would be useful to have a small table that details the finalised conditions / rna buffer that one would select when next needing to send huMPCs to the ISS.

Thank you

Author Response

We wish to thank the Reviewer for her/his overall positive evaluation of the manuscript, and for her/his suggestions that enabled us to improve the manuscript. Here is our point-to-point reply to the Reviewer comments.

  1. My only minor comment is the combination of graphs (figure 4, figure 5) to reduce whitespace and improve comparison between GM/DM and Plate/EU.

Based on the Reviewer suggestion, we have reorganized the graphs in Figures 4 and 5 to allow easy comparison between huMPCs cultured in plates and EUs in GM or DM conditions. 

  1. Increase in contrast of images for figure 2, and in greyscale, would improve visibility of cells (like figure 3)

We imagined the Reviewer was referring to Figure 1. We have converted this figure in gray scale and contrasted it for a better visibility, as suggested.

  1. Finally, it would be useful to have a small table that details the finalised conditions / rna buffer that one would select when next needing to send huMPCs to the ISS.

We have added the suggested table (Table 5 in the revised manuscript).